# Caution Is Needed When Using Niche Models to Infer Changes in Species Abundance: The Case of Two Sympatric Raptor Populations

**DOI:** 10.3390/ani11072020

**Published:** 2021-07-06

**Authors:** Adrián Regos, Luis Tapia, Alberto Gil-Carrera, Jesús Domínguez

**Affiliations:** 1Departamento de Zooloxía, Xenética e Antropoloxía Física, Universidade de Santiago de Compostela, Campus Sur, 15782 Santiago de Compostela, Spain; luis.tapia@usc.es (L.T.); jesus.dominguez@usc.es (J.D.); 2CIBIO/InBIO, Research Center in Biodiversity and Genetic Resources, Predictive Ecology Group, Campus Agrario de Vairão, R. Padre Armando Quintas, N° 7, 4485-661 Vairão, Portugal; 3EBX, Estación Biolóxica do Xurés, Vilameá 121, 32870 Lobios, Spain; atoupa@hotmail.com; 4GREFA, Monte del Pilar S/N, 28220 Majadahonda, Spain

**Keywords:** mountain heathlands, open-habitat raptors, habitat suitability models, species abundance–habitat suitability relationship, population changes

## Abstract

**Simple Summary:**

This study focuses on sympatric populations of hen harrier (*Circus cyaneus*) and Montagu’s harrier (*Circus pygargus*) breeding in mountain heathlands in the NW Iberian Peninsula. These populations have been affected by habitat loss caused by land-use change. Despite the mounting evidence supporting positive relationships between species abundance and habitat suitability, the capacity of ecological niche models to capture variations in population abundance remains largely unexplored. This study shows that habitat suitability predicted from niche models was significantly correlated with the relative abundance for hen harrier and, to a lesser extent, for Montagu’s harrier. However, the temporal variation in local population abundance was not significantly explained by habitat suitability changes predicted by the niche models. These findings call for caution in the use of niche models to infer changes in population abundance. The positive relationship between species abundance and habitat suitability supports the use of niche models to estimate abundance but does not guarantee the ability of these models to predict temporal variations in species abundance. These findings highlight (1) the need to include other possible biotic or abiotic factors involved in population abundance dynamics into niche models and (2) the need to establish specific monitoring protocols for tracking population dynamics.

**Abstract:**

Despite the mounting evidence supporting positive relationships between species abundance and habitat suitability, the capacity of ecological niche models (ENMs) to capture variations in population abundance remains largely unexplored. This study focuses on sympatric populations of hen harrier (*Circus cyaneus*) and Montagu’s harrier (*Circus pygargus*), surveyed in 1997 and 2017 in an upland moor area in northwestern Spain. The ENMs performed very well for both species (with area under the ROC curve and true skill statistic values of up to 0.9 and 0.75). The presence of both species was mainly correlated with heathlands, although the normalized difference water index derived from Landsat images was the most important for hen harrier, indicating a greater preference of this species for wet heaths and peat bogs. The findings showed that ENM-derived habitat suitability was significantly correlated with the species abundance, thus reinforcing the use of ENMs as a proxy for species abundance. However, the temporal variation in species abundance was not significantly explained by changes in habitat suitability predicted by the ENMs, indicating the need for caution when using these types of models to infer changes in population abundance.

## 1. Introduction

Habitat loss is the most important threat to biodiversity worldwide and strongly impacts the distribution and abundance of both animal and plant species [1]. “Species distribution” and “population abundance” are considered essential biodiversity variables (EBVs) by the Group on Earth Observations Biodiversity Observation Network (GEO BON) [2]. Long-term time series of population abundance (obtained from standardized monitoring protocols involving repeated surveys of the same population) are essential for population management but very costly and time-consuming [3].

Model-assisted monitoring has recently emerged as an alternative, cost-effective way of assessing the impact of environmental change on biodiversity [4]. Ecological niche models (also known as habitat suitability or species distribution models [5]), which correlate species occurrence data and current environmental covariates (mostly climatic and topographic variables), have been widely used by ecologists in recent decades [6]. Species occurrence data are far more common than species abundance data, which require more costly and time-consuming sampling methods. Consequently, abundance data are often substituted by environmental suitability predicted by ecological niche models (ENMs), by assuming positive relationships between species abundance and habitat suitability [7,8]. The use of ENMs to produce data that can be used as a proxy for species abundance has been supported by many studies over the last years (see [9] and references therein). Despite the mounting evidence supporting such relationships, other studies have also reported weak (or no) correlation between species abundance and environmental suitability predicted by ENMs [10], placing into question the capacity of such models to capture the variation in species abundance. A common limitation of niche models is that they often relied on climatic envelopes inferred from macroclimatic datasets [9], therefore resulting in predictions of habitat suitability geographically overestimated. This limitation highlights the need to investigate nonclimatic sources of population abundance variation at local scales [10].

According to the “Habitat Selection Theory”, individuals will select habitats that maximize their biological “fitness”. The importance of local habitat conditions has been corroborated in several studies in which variables other than climate-related variables are used to determine environmental suitability [10]. In this regard, satellite remote sensing (SRS) provides continuous and standardized measures of the environment at high spatial resolution, allowing cost-effective and reliable characterization of species habitats at local scales [11,12]. Thus, SRS data have increasingly been used as predictor variables for both species distribution and abundance models in recent years [13,14]. For instance, the incorporation of SRS variables related to water and carbon cycles (e.g., the normalized difference water and vegetation indices (NDWI and NDVI, respectively)) has recently been reported to be critical for model-assisted monitoring of endangered plant and animal species [13,15]. These studies suggest that calibrating ENMs with SRS variables would enable the accurate prediction of species distributions at local scales (across space and time). In light of the habitat selection theory, we would expect model-predicted environmental suitability to be significantly correlated with species abundance, thus supporting the use of SRS-based ENMs to estimate species abundance. If so, both the spatial and temporal variation in species abundance should be explained (at least partly) by changes in environmental suitability predicted by these models. 

This study focuses on sympatric populations of hen harrier (*Circus cyaneus*) and Montagu’s harrier (*Circus pygargus*) breeding in mountain heathland (Natura 2000 habitat type: 4020 European wet heaths, 4030 European dry heaths and 7110 active raised bogs listed in Annex I of the Habitat Directive 92/43/EEC) in the NW Iberian Peninsula [16,17]. Both species are included in the Annex I of the European Bird Directive with conservation status “Vulnerable” in Spain (see [18]). These species are considered good indicators of environmental change due to their high susceptibility to being affected by habitat loss and fragmentation caused by land-use change [19]. In fact, the populations have declined sharply in recent years in Spain, especially in the northwest of the country, due to intensive agroforestry and land-use change [18,20,21]. In particular, the habitat availability and quality of both target species have markedly decreased in our region over the last 20 years due to large-scale land-use conversions from mountain heathlands and grasslands to fast-growing tree plantations (mostly *Pinus* and *Eucalyptus* spp.) and native forest expansion caused by intensive forest practices and rural abandonment processes (i.e., loss of traditional agricultural and livestock practices) [14,20,21].

To test the above-mentioned hypotheses, we assessed the predictive accuracy of ENMs (i.e., realized environmental space *sensu* “Grinnellian niche” [22,23,24]) based exclusively on SRS variables (derived from Landsat satellite imagery) and occurrence records (in situ data) and quantified the extent and direction of changes in environmental suitability predicted by these models for each species over the last 20 years (between 1997 and 2017). Landscape change analysis was also performed using the same Landsat satellite images to clarify the underlying processes affecting species habitat changes. Finally, we tested whether the environmental suitability predicted by the ENMs is significantly correlated with species abundance and thus supports the use of RS-based ENMs for estimating abundance and whether the spatiotemporal variation in species abundance is explained by changes in environmental suitability predicted by these models.

## 2. Materials and Methods

### 2.1. Study Area

The study area is located in the “Serra do Laboreiro”, which is included in the “Baixa Limia” Site of Community Importance (SCI) and which is a Special Protection Area for birds (SPA) (Galicia, NW Spain) (Figure 1). The study site covers an area of 3350 ha, bordering to the north with the Peneda-Gerês National Park (North of Portugal). It is a mountain range, with an average elevation of around 800 m a.s.l., a gradual, gentle relief and summits of up to 1300 m, predominantly comprised of granite rock. The climate in the area is temperate sub-Mediterranean oceanic, with an annual mean temperature of 8–12 °C and mean annual precipitation of 1200–1600 mm [25]. The landscape is representative of European Atlantic mountain heathlands, dominated by upland moors and open vegetation communities, mainly European heaths (*Erica* sp., *Chamaespartium tridentatum*, *Ulex* sp., *Cytisus* sp., among others), with a strong influence of extensive livestock grazing [26]. Mountain heathlands are nowadays under threat and very restricted in southern Europe, being priority conservation areas in Spain [27,28]. Forests are fragmented and dominated by pine (*Pinus sylvestris*) in the highest areas and oaks (*Quercus robur*, *Q. pyrenaica*) at lower elevations and in valleys [29]. The human population in the area is currently very sparse, although the landscape has been intensely managed by humans for centuries and is especially associated with the presence of extensive livestock. Rural communities still use fire to create pastures for extensive livestock, which leads to frequent intentional wildfires in the area, mainly at the end of winter.

### 2.2. Field Surveys and Target Species

The hen harrier and Montagu’s harrier populations were surveyed every month during the 1997 breeding season, between 15 April and 31 August. Despite the migratory behavior of the Montagu’s harrier and dispersive movements of the hen harrier, these populations are very philopatric (i.e., individuals tend to return to breed in this area every year) so no individual was recorded in migration. In our study, the fledglings stayed in the breeding area until September (no migratory restlessness was observed). The survey was repeated in 2017 with the same timing as in the 1997 census to minimize any phenological bias. In each breeding season, two road “point transect censuses” [30] were carried out from a 4 × 4 vehicle, taking advantage of the vast unpaved track network in the study area (see Figure 1). Both road censuses were 9 km long, and point counts were established at intervals of 1 km to avoid pseudoreplication, with a total of 10 per transect. All visual sightings of individual birds were recorded during 10 min in each point count. In each transect, every individual was controlled to avoid double counts. Two experts in raptor identification carried out all censuses in both years. The precise locations of observations were recorded via a Garmin GPS handheld receiver (accuracy of 15 m ± 3 m). Finally, 20 point counts were conducted every month (*n* = 100 per year), yielding a total of 1000 min of effective sampling effort per year. The surveys revealed a decrease in the total number of hen harrier recorded between 1997 and 2017 (*n*_1997_ = 13; *n*_2017_ = 6), but no changes in numbers of Montagu’s harrier (*n*_1997_ = 11; *n*_2017_ = 11).

### 2.3. Satellite-Based Land Cover Mapping

We used freely available Earth observation (EO) imagery to derive land use and land cover (LULC) maps for 1997 and 2017. The main EO data source consisted of optical multispectral bands (30 m resolution) from two Landsat images acquired on dates as close as possible to the field survey. More specifically, we used spectral bands from Landsat 5 TM (5 April 1997) and Landsat 8 OLI sensors (28 April 2017). All scenes were downloaded from the United States Geological Survey (USGS) Global Visualization Viewer (http://glovis.usgs.gov, accessed on 1 November 2020). All Landsat scenes were processed to Standard Terrain Correction (Level 1T), which provides systematic geometric accuracy by incorporating ground control points while applying a digital elevation model (DEM) for topographic accuracy. Digital numbers (DNs) were converted to top-of-atmosphere radiance and physically meaningful units by radiometric calibration and application of sensor- and band-specific calibration parameters. The classification process was based on the radiometric information obtained from reflective bands and two multispectral indices for each image: (1) the normalized difference vegetation index (NDVI [31]) and (2) the normalized difference water index (NDWI [32]), to enhance the contribution of vegetation in the spectral response and mitigate other factors such as soil, topography, lighting conditions and atmosphere [31]. Topographic information was obtained from NASA’s Shuttle Radar Topography Mission (GDEM-SRTM) (https://www2.jpl.nasa.gov/srtm/ accessed on 1 November 2020).

Supervised classification was carried out using the following four classification algorithms available in the R-based package “Caret” and implemented in the “RStoolbox” package, version 0.1.534: (1) stochastic gradient boosting, (2) random forest, (3) support vector machines (SVMs) with linear kernel and (4) SVMs with radial basis function kernel. The final land use/cover maps were elaborated by applying an ensemble approach, including the simple voting system (also known as “majority voting” or “select all majority” system, *sensu* [33]) that includes those with an overall accuracy higher than 95% (see Appendix A). The four most important land use and vegetation cover types identified during the fieldwork were: (1) farmland (extensive agricultural areas interspersed with native deciduous forest, mostly oak woodlands), (2) open heathland (open vegetation areas dominated by dry heaths and largely affected by burning and extensive livestock rearing), (3) closed heathland (closed/mature heath areas dominated by dry and wet heaths, including peat bogs) and (4) pine plantations. Training and validation areas for each LULC class were established by on-screen digitizing, with QGIS software, and consisted of a set of pixels identified over well-known homogeneous areas in each Landsat image, thus providing a reference spectral signature for each class. Once the ensemble LULC maps were constructed, net changes were quantified for each land cover type for the entire study period (1997–2017) in the area, including 500 m buffers around each point count (see Table 1).

Data importation, preprocessing, computation of spectral indices and image classification were performed using the RStoolbox package, version 0.1.534. Land-use/cover change analysis was performed with the “lulcc” v.1.0.2 package in R [34].

### 2.4. Ecological Niche Modelling

To quantify changes in habitat suitability at the landscape level for both species between 1997 and 2017, we first constructed ecological niche models (ENMs; i.e., realized environmental space *sensu* “Grinnellian niche” [22,23,24]). These ENMs allowed us to empirically correlate hen harrier occurrence data (presence and absence data registered by point count for each year of sampling) and the percentage of each LULC class (pine plantations, farmland, open and closed heathlands), topography (altitude, slope and aspect) and water content of vegetation (computed from the NDWI) within a radius of 500 m around each point count. We applied eight widely used modeling algorithms implemented in BIOMOD2 to deal with the uncertainty of different modeling techniques and to provide more informative and ecologically correct predictions [35]: (1) generalized linear models, (2) flexible discriminant analysis, (3) classification tree analysis, (4) multivariate adaptive regression splines, (5) maximum entropy, (6) random forest, (7) artificial neural networks and (8) surface range envelope [36]. We then tested the performance of the different models by splitting the original raptor data set into two subsets: 70% of the data was used to train the models and the remaining 30% was used to validate the model. We randomly repeated this procedure 30 times to produce predictions independent of the training data. We applied a weighted average approach to compute a consensus of single-model projections by using the area under the receiver operating characteristic (ROC) curve (AUC) to weight the model [37]. Only models with AUC values above 0.8 were used in the ensemble procedure. The ensemble models, obtained from the consensus of single models, were directly projected at 60 m resolution [38]. We calculated two different evaluation indices: the area under the ROC curve (AUC) [39] and true skill statistic (TSS) [40].

### 2.5. Testing the Species Abundance–Habitat Suitability Relationship

To test the hypotheses about the relationship between species abundance and habitat suitability (see Section 1), we related the relative abundance index (estimated as the number of individuals recorded in each point count and sampling year) to the habitat suitability (hereafter HS) predicted by the ENMs for each species and year. For this purpose, we fitted generalized linear models with a negative binomial distribution of errors and logarithmic link function with the function “glm.nb” available in the R package “MASS”, version 7.3–51.1. We also explored the effect of habitat changes predicted by ENMs between 1997 and 2017 on the variation in species abundance (estimated as the difference in the abundance index between 1997 and 2017) by fitting linear models. In addition, we used Pearson’s correlation coefficient to test for any association between species abundance and HS, and we used Spearman’s rho to test for any relationship between variation in species abundance and change in HS. For the sake of caution, only effects with *p*-values < 0.01 were considered significant.

## 3. Results

Overall, the ensemble ENMs performed very well for both species, as measured by AUC (AUC hen harrier = 0.95 and AUC Montagu’s harrier = 0.93) and TSS metrics (TSS hen harrier = 0.85 and TSS Montagu’s harrier = 0.75). In relation to the importance of the environmental variables included in niche models, the results reflect the ecological requirements of both species (Figure 2). As expected, the occurrence of both species was mainly correlated with open and closed heathlands (hunting and breeding areas, respectively), although the NDWI was only found to be important for hen harrier, indicating its preference for areas with damp vegetation (wet heaths and peat bogs).

The types of land-use change detected have led to contrasting changes in the habitat availability for both species over the last 20 years (Table 2, Figure 3). Thus, the ENMs predicted a marked increase in the habitat availability for Montagu’s harrier and a significant reduction for hen harrier (Table 2, Figure 3).

Regarding the land-use changes within areas indicated by the ENMs to be suitable habitat in both years, the results showed no important changes, as these areas are closely associated with open heathland for both species but also with farmland in the case of hen harrier (see “Overlap” in Figure 4). Areas where raptor habitat increased between 1997 and 2017 were associated with important conversion from closed to open heathland for both raptor species and to farmland for hen harrier (see “Gains” in Figure 4). Finally, areas with predicted habitat loss were linked to extensive conversion from open to closed heathland and pine plantations (see “Losses” in Figure 4).

A significant positive correlation between abundance and habitat suitability indices was obtained for both species and years (Figure 5A). In particular, the results showed that habitat suitability predicted by the ENMs significantly explained the observed abundance of hen harrier (pseudoR^2^_Nagelkerke_ = 0.373; *p*-value < 0.001) and, to a lesser extent, that of Montagu’s harrier (pseudoR^2^_Nagelkerke_ = 0.271; *p*-value < 0.01). The results also showed that the greater the increase in habitat suitability, the greater the increase in observed abundance (Figure 5B). However, the positive relationships between species abundance and habitat suitability were not strong enough to explain changes in species abundance (R^2^ = 0.1379; *p*-value = 0.036).

## 4. Discussion

Our findings support the species abundance–habitat suitability hypothesis, reinforcing the use of ecological niche models (ENMs) as a proxy for species abundance. However, the temporal variation in species abundance was not significantly explained by changes in habitat suitability predicted by the ENMs, which calls for caution in the use of ENMs when inferring changes in population abundance.

Various authors have already highlighted the positive relationships between habitat suitability and abundance/distribution patterns [8,9]. As both distribution and abundance patterns, particularly for raptors, are determined by the availability of suitable nesting places and trophic resources, habitat characteristics should *a priori* be good predictors of abundance [41]. However, the use of abundance and distribution data to infer relationships between species and habitat characteristics should be considered with caution, as these data can misleadingly reflect habitat suitability and actual carrying capacity [42,43]. The concept of carrying capacity encapsulates the notion that habitat resources must ultimately limit the numbers of raptors in an area. Nevertheless, factors other than those purely related to habitat characteristics (e.g., human pressure, intra- and interspecific interactions, period of the annual cycle) are known to affect local abundance and population dynamics [44,45]. Although our findings showed that species abundance and habitat suitability are significantly correlated, they also indicated that changes in abundance between 1997 and 2017 are not exclusively determined by changes in habitat suitability. Therefore, other variables, not directly associated with habitat dynamics, may have affected the observed changes in abundance. For instance, direct persecution (illegal killing), human-related disturbance of nest and foraging areas, predation and the effect of pesticides are typical examples of relevant factors that cannot be easily incorporated in ENMs [46,47]. In addition, testing the relationship between abundance and habitat suitability requires a good estimation of the optimal environmental conditions for the species (see [48,49,50]). This implies the use of ENMs that are better at characterizing the fundamental niche of the species than those based exclusively on local habitat variables (i.e., more related to the realized niche of species). Moreover, modeling endangered species (i.e., species often with restricted distributions) is particularly challenging since ENMs are often built from few records and at local scales, which affects both model calibration and evaluation procedures [51,52,53]. Moreover, the semicolonial and migratory behaviors of species such as Montagu’s harrier or the large hunting distances of breeding male and female hen harriers [54,55] can reduce the ability of ENMs to infer changes in abundance since surveys are more prone to double counts or pseudoreplication than for other sedentary and colonial species. Another issue that can constrain our ability to infer population dynamics from ENMs is the interannual fluctuations in prey availability. In this regard, the inclusion of remotely sensed ecosystem functional variables (e.g., annual mean of NDVI or NDWI as a proxy for primary productivity [56], food resources [57] or prey availability [58]) into ENMs has been found to improve model predictions [13,15,58,59].

Despite these limitations, our findings suggest an important effect of habitat quality on the patterns of distribution and abundance for both raptor species. The ENMs provided good fits to the data for both species (with AUC and TSS values of up to 0.95 and 0.85, respectively), which allowed us to test with confidence whether ENM-predicted habitat suitability was correlated positively with species abundance. The ranked importance of the environmental variables included in ENMs satisfactorily reflected the ecological requirements of the target species in the study area [16,17]. As expected, both species were mainly associated with open and closed heathland (hunting and breeding areas, respectively). Optimal habitat conditions for these species are represented by a mosaic of heather and grassland [17]. Areas dominated by tall heather, particularly those furthest from forest harvesting areas, are the most valuable for hen harrier nesting/roosting. Sites with little or no heather (other than very wet blanket bogs) or completely dominated by tall woody heather may need management interventions [60]. More interestingly, our findings highlight the importance of vegetation water content (i.e., wet heathland and peat bogs) in explaining the distribution of hen harrier, which to our knowledge is one aspect of the ecological niche not documented to date in the Iberian Peninsula, although known in other more northern areas of Atlantic Europe [60,61]. This finding highlights the usefulness of remotely sensed descriptors of heathland water balance for characterizing the species niche and confirms the need to go beyond the traditional habitat description (i.e., often based on landscape structural and compositional attributes) [13,58].

Land-use change has led to changes in habitat availability for both species (Table 1 and Table 2). The ENM projections revealed a large decrease in the habitat available for hen harrier and an overall increase in suitable habitat for Montagu’s harrier. Areas where raptor habitat changed between 1997 and 2017 were associated with important changes from mature/closed to open heathland for both species and to farmland for hen harrier. Heath and shrub formations in the mountainous areas of NW Spain have historically been linked to extensive agropastoral activities. However, these extensively managed systems have gradually been abandoned and largely replaced by monoculture forest plantations (mostly of Scots pine *Pinus sylvestris* in the study area) and semi-intensive grassland [27,28,29]. Side-effects on these raptor populations are traditionally associated with the conservation status of wet heath and peat bog moorland in Atlantic Europe [60,61,62], but these habitats are scarce and relict in the context of the Iberian Peninsula [27,28]. Furthermore, our findings suggest that forest plantations for timber production have limited the foraging opportunities and reproductive fitness, with potential impacts at population scale [60]. Our findings also showed a large reduction in habitat availability for hen harrier associated with the loss of wet heathland and peat bogs in the study area (Appendix A).

The Common Agricultural Policy (CAP) and European Regional Development Funds (FEDER Funds) aim to maintain traditional agropastoral systems to ensure the conservation and sustainable management of these priority habitats. However, these EU policies are failing in regard to biodiversity and land degradation [63]. Urgent action is required to achieve a sustainable model for European agriculture that can jointly benefit biodiversity and rural communities in mountain heathlands [63,64].

## 5. Conclusions

Our ecological niche models performed very well for both raptor species, predicting a marked increase in the habitat availability for Montagu’s harrier and a significant reduction for hen harrier. In relation to the ranked importance of the environmental variables included in the niche models, our findings identified the ecological requirements for both species. The presence of both species was mainly correlated with open and closed heathlands, although the normalized difference water index was found to be the most important for hen harrier, indicating the preference of this species for wet heathland and peat bog areas. Our findings showed that habitat suitability predicted from niche models was significantly correlated with the relative abundance for hen harrier and, to a lesser extent, for Montagu’s harrier (due likely to its migratory and semicolonial behavior). However, the temporal variation in local population abundance was not significantly explained by habitat suitability changes predicted by the niche models. These findings call for caution in the use of niche models to infer changes in population abundance. The positive relationship between species abundance and habitat suitability supports the use of niche models to estimate abundance but does not guarantee the ability of these models to predict temporal variations in species abundance. These findings highlight (1) the need to include other possible biotic or abiotic factors involved in population abundance dynamics into niche models and (2) the need to establish specific monitoring protocols for tracking population dynamics. The inherent difficulties associated with the migratory and semicolonial behavior of the Montagu’s harrier together with the large hunting distances of breeding male and female hen harriers highlight the need for sampling protocols supported by marked individuals (e.g., with radio tracking) that allow tracking their habitat occupancy along the annual cycle or improving ENMs by including time series of remotely sensed variables as proxies for interannual habitat dynamics (e.g., annual variations in prey availability).

## Figures and Tables

**Figure 1 animals-11-02020-f001:**
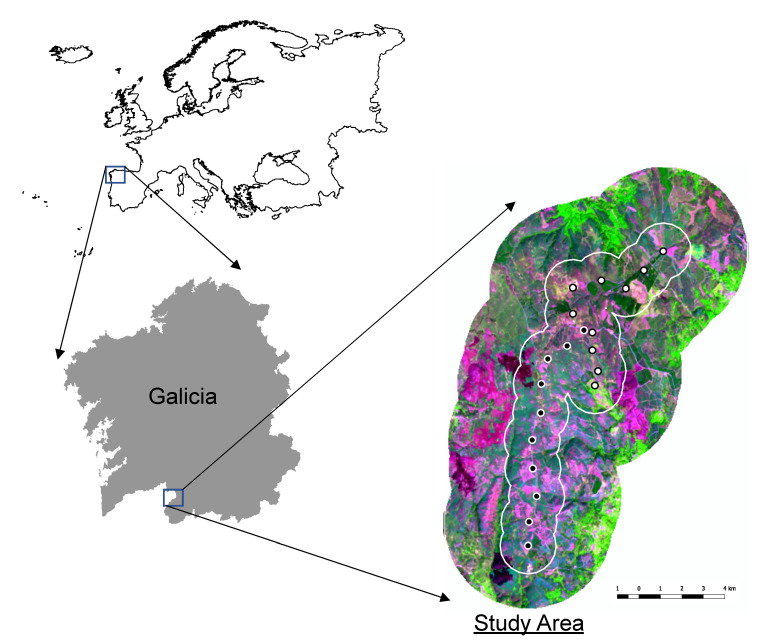
Location of the study area in the “Serra do Laboreiro”. False-color RGB composite from Landsat 8 OLI sensor imagery (white and black lines represent 1 km and 2 km buffer areas around each point count, respectively; white and black circles represent each transect).

**Figure 2 animals-11-02020-f002:**
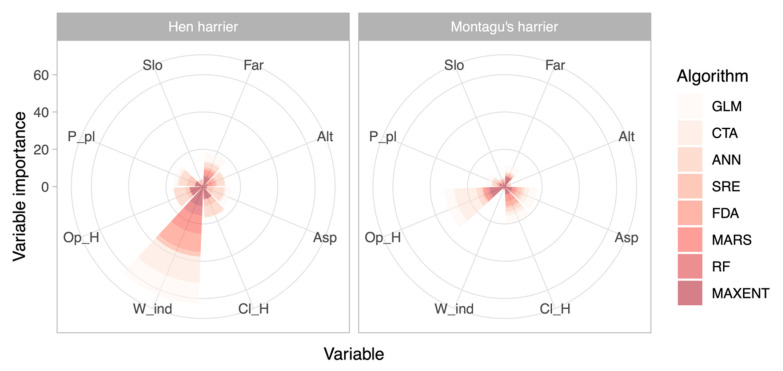
Ranked importance of the environmental variables included in ecological niche models of hen harrier and Montagu’s harrier. Values represent the importance of each variable accumulated from the different modeling techniques. GLM, generalized linear models; CTA, classification tree analysis; ANN, artificial neural networks; SRE, surface range envelope; FDA, flexible discriminant analysis; MARS, multivariate adaptive regression splines; RF, random forest; MaxEnt, maximum entropy; Slo, slope; Far, farmland; Alt, altitude; Asp, aspect; Cl_H, closed heathland; W_ind, normalized difference water index; Op_H, open heathland; P_pl, pine plantations.

**Figure 3 animals-11-02020-f003:**
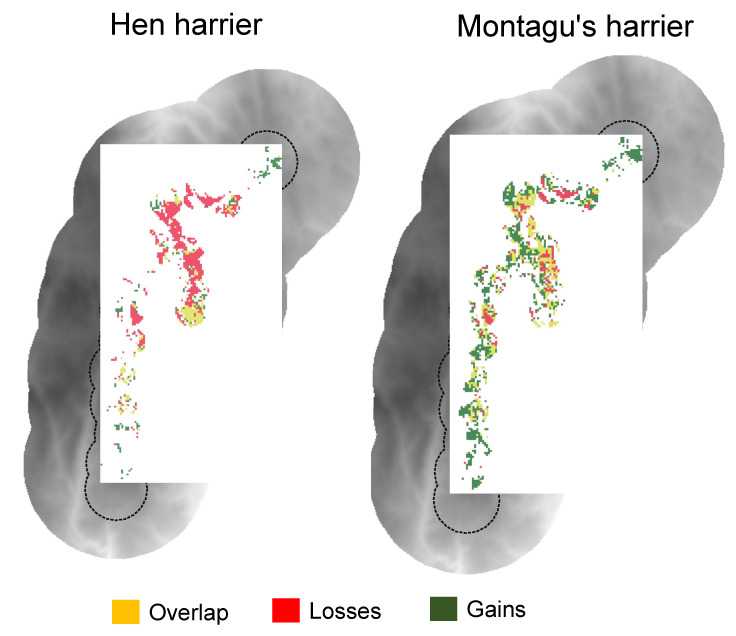
Maps depicting areas predicted by the ecological niche models as suitable in 1997 and 2017 (“Overlap”), only suitable in 1997 (“Losses”) and only suitable in 2017 (“Gains”).

**Figure 4 animals-11-02020-f004:**
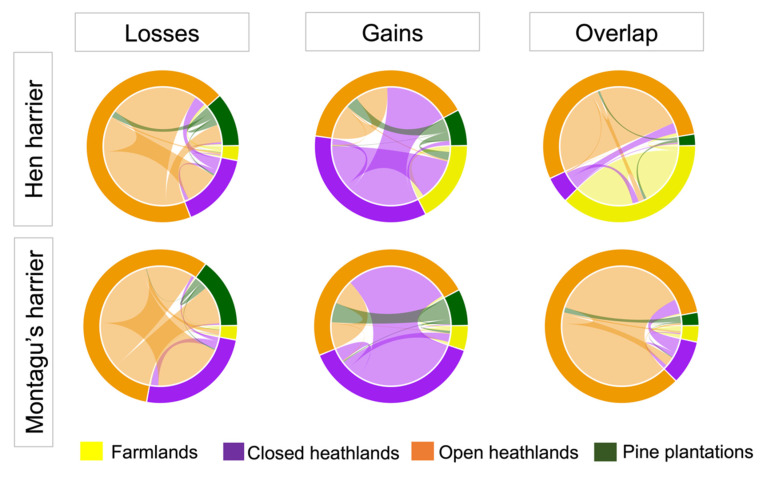
Circular plot illustrating the changes in land cover between 1997 and 2017 within areas predicted by the ecological niche models as suitable habitat in 1997 and 2017 (“Overlap”), only suitable in 1997 (“Losses”) and only suitable in 2017 (“Gains”). The width of the lines is proportional to the contribution of each type of land type to the change. The different colors represent the different types of land cover.

**Figure 5 animals-11-02020-f005:**
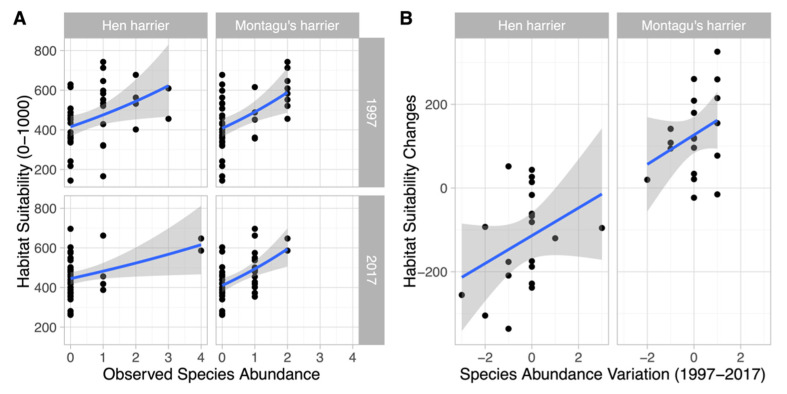
The species abundance–habitat suitability relationship: (**A**) degree of correlation between observed species abundance and habitat suitability predicted from ENMs for each species and year (Montagu’s harrier (Spearman’s ρ = 0.57; *p*-value < 0.001) and hen harrier (Spearman’s ρ = 0.55; *p*-value < 0.001)); (**B**) degree of correlation between species abundance variation and habitat suitability changes predicted from ENMs for each species between 1997 and 2017 (hen harrier (Pearson’s r = 0.34; *p*-value = 0.12) and Montagu’s harrier (Pearson’s r = 0.31; *p*-value = 0.17)).

**Table 1 animals-11-02020-t001:** Landscape change (ha) within 500 m buffer around each point count.

	1997	2017
Farmland	47.79	83.07
Closed heathland	728.19	388.08
Open heathland	402.57	695.97
Pine plantations	152.82	164.25

**Table 2 animals-11-02020-t002:** Habitat availability (ha) predicted by ecological niche models for 1997 and 2017.

	1997	2017
Hen harrier (*Circus cyaneus*)	202.77	121.32
Montagu’s harrier (*Circus pygargus*)	140.22	258.48

## Data Availability

Landsat images are freely available from the United States Geological Survey (USGS) Global Visualization Viewer (http://glovis.usgs.gov accessed on 1 November 2020). Topographic information was obtained from NASA’s Shuttle Radar Topography Mission (GDEM-SRTM) (https://www2.jpl.nasa.gov/srtm/ accessed on 1 November 2020). Raptor data are available from the corresponding author on reasonable request.

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
