# Peer review of "Caution Is Needed When Using Niche Models to Infer Changes in Species Abundance: The Case of Two Sympatric Raptor Populations"

_animals, 2021, doi:10.3390/ani11072020_

Round 1
Reviewer 1 Report
General comments
The manuscript is consistently very well written, both in terms of language and in terms of content!
However, that the temporal variation in species abundance could not be significantly explained by changes in habitat suitability predicted by the ENMs is indeed remarkable, when considering the dramatic change of open and closed heathland cover, the two variables which proved being extremely important predictors in the calculated ENMs. Hence, I have some doubts as to whether the data is of sufficient quality to reliably examine the questions addressed, for the following reasons:
(1) The study area is pretty small (only 33.5 km2) for a study on raptors. Can the authors provide an estimate on the number of breeding pairs of both species, which bred in the area in individual years? When most of the data points are generated by only few breeding pairs, pseudoreplication may be a substantial problem!
(2) A further emerging problem may be that although breeding Hen Harriers predominately hunt within 1-2 km from their nest, they can travel up to 9 km from their nests and can have an average home-range of 8 km2 (Arroyo et al. 2014, Bird Study 61/1). Hence, it is very likely that the same birds were counted at various census points. Again, then pseudoreplication / spatial autocorrelation may severely weaken the reliability of the models.
(3) In Montagu’s harrier the situation may be even more problematic since this species is semi-colonial (e.g. Arroyo et al. 2004, BWP Update 6: 41-55). This may explain why predictions of habitat suitability based on occurrence data do not result in close correlations with spatio-temporal abundance changes.
At least it has to be comprehensively addressed in the discussion to what extent these two issues (2) and (3) may affect the performance of the calculated models!
(4) Surveys were conducted between 1 April and 31 August. Since April is the main spring migration period of Montagu’s harrier many migrants may pass the area, which may utilize different habitats than breeding birds. Combining both types of birds – breeding and migrating individuals – may affect the predicted habitat suitability. This is an issue, which should be also clarified.
Specific comments
It is unnecessary in the text of the discussion to refer to specific figures again in parentheses.
Figure 2: I fear that in this black-and-white figure not everything can be seen that is necessary for the interpretation of the results. Please check.
Academic Editor Notes
I have now received reports from the two reviewers for your ms, and I have read the ms by myself. One reviewer has given only minor comments, whereas the other (R1) has pointed some major concerns related on your method/data quality. I agree with the R2. Therefore, your ms will need a major revision before I can consider it for the publication for the Animals.
Dear Editor, Jukka Jokimäki,
We are pleased to submit a revised version of the manuscript ‘1265920’, entitled “Caution is needed when using niche models to infer changes in species abundance: the case of two sympatric raptor populations”. First, we would like to thank you for handling our manuscript and your personal and careful revision. We are also grateful to the two reviewers, whose comments and suggestions helped us to improve the quality of our manuscript.
In this revised version, we addressed thoroughly all comments made by the Editor and reviewers. Specifically, regarding the reviewers’ major concerns, we have better justified the suitability of the survey methods and discussed potential limitations related to the migratory and semi-colonial behavior of the Montagu’s harrier. Still, we have also explained that pseudoreplication has been considered during the surveys and why is not an issue of special concern in our case. We have also discussed other limitations such as the interannual variation in prey availability and how this could be addressed in future studies. The remaining minor issues have also been addressed and/or clarified in the revised manuscript.
In ‘response to reviewers' comments’, we provide our point-by-point responses to all comments or suggestions raised by the reviewers, indicating the lines where changes have been included in the revised version of the manuscript. The clean version of the revised manuscript is tagged as ‘Clean Version’. Changes are highlighted by using the ‘tracked changes mode’ in MS Word, after the clean version. In the detailed point-by-point responses we refer to the line numbers of the “clean” version of the manuscript for an easier reading.
Please note that Figures 2, 4 and 5 were wrong in the PDF of the original submission (not in the Word file). It might have been caused by the automatic conversion from Word to PDF during the submission process. We apologize for this error. We want to ask the Editorial team for the possibility of uploading the figures as separate files to ensure a correct visualization.
The figures (in pdf and tiff) are in the zip file but also in this personal link:
https://nubeusc-my.sharepoint.com/:f:/g/personal/adrian_regos_usc_es/Eg2jXBBLDMROt4kOA8BvzxYB7xq6QwzQGQ2EOiYvt9SNSg?e=A0hKxG
We hope that this revised version of the manuscript meets the journal’s expectations for publication.
We look forward for your editorial opinion.
Sincerely,
Adrián Regos on behalf of all co-authors.
----------------------------
The major concerns are related to:
1) The study area is pretty small (only 33.5 km2) for a study on raptors
Response: The study area is small but very representative of the European Atlantic mountain heathlands that these species has historically used to breed in NW Spain. Unfortunately, mountain heathlands are nowadays under threat and very restricted in southern Europe, being priority conservation areas in Spain (Regos et al 2013; Fagúndez et al 2013). This area is included in a Natura 2000 site, being one of the core breeding areas of these two species in our region. We have now remarked in the revised version the representativeness of our study area in the European context to help the audience (including researchers working with these species) understand the relevance of our results for other Atlantic Mountain heathlands (lines 140 and 144-145); and two new references (highlighted in yellow):
Rego, P. R.; Guitián, M. A. R.; Castro, H. L.; Da Costa, J. F.; Sobrino, C. M. Loss of european dry heaths in NW Spain: A case study. Diversity 2013, 5, 557–580.
Fagúndez, J. Heathlands confronting global change: Drivers of biodiversity loss from past to future scenarios. Ann. Bot. 2013, 111, 151–172.
In addition, we think that the fine-scale habitat characterization based on Landsat images (30-m resolution) allows to model these species with confidence. In fact, land-use changes and their impacts on habitat quality predicted by our models are in line with other studies carried out in our region at larger scale. See our previous studies:
Regos, A.; Tapia, L.; Gil-Carrera, A.; Domínguez, J. Monitoring protected areas from space: A multi-temporal assessment using raptors as biodiversity surrogates. PLoS One 2017, 12, e0181769.
Tapia, L.; Dominguez, J.; Rodriguez, L. Modelling habitat selection and distribution of Hen harrier (Circus cyaneus) and Montagu’s harrier (Circus pygargus) in a mountainous area in Galicia. J. Raptor Res. 2004, 38, 133–140. 441
Tapia, L.; Domínguez, J.; Rodríguez, L. Hunting habitat preferences of raptors in a mountainous area (northwestern Spain). Polish J. Ecol. 2008, 56, 323–333.
Tapia, L.; Regos, A.; Gil-Carrera, A.; Dominguez, J. Assessing the temporal transferability of raptor distribution models: Implications for conservation. Bird Conserv. Int. 2018, 28, 375–389. 450
Tapia, L.; Regos, A.; Gil-Carrera, A.; Domínguez, J. Unravelling the response of diurnal raptors to land use change in a highly dynamic landscape in northwestern Spain: an approach based on satellite earth observation data. Eur. J. Wildl. Res. 2017, 63, 40.
2) There is a possibility of the pseudoreplication
Response: the observers (which are also co-authors of the work) has a vast experience (more than 20 years) working with raptors in our region. We are aware of the implications of pseudoreplication, so this issue has been carefully considered during the sampling protocols. Within each transect every individual was controlled to avoid doble counting. The advantage of our protocol is that every transect was carried out several times over the breeding period to ensure that all absences are “real” and not “pseudo-absence” due to an insufficient sampling effort. If several observations are recorded in one sampling point across the breeding period, the record remains as “presence”. On the contrary, if after all surveys there were no observations, it would be recorded as “absence”. We think that our sampling protocol allows to be confidence about the habitat characterization and preference of this population in our study area. However, following the Editor’s and reviewer#1 suggestions, we now discuss potential problems associated with pseudo-replication (see lines 370 and 373).
3) One of your study species, Montagu’s harrier, is semi-colonial
At least it has to be comprehensively addressed in the discussion to what extent these three issues have been tried to controlled during the field methods, and how these topics may affect the performance of the calculated models!
Response: We agree that this issue deserves to be mentioned for the case of Montagu’s harrier for readers that are not familiar with these species. The observers have a deep knowledge of the ecology of both species. The semi-colonial behavior of the Montagu’s harrier was considered during the transect by controlling all individuals recorded at each point. The probability of double counting the same individual during the transect was minimum (now clarified in line 167). However, the semicolonial behavior of some species is in fact a very interesting issue that has not been mentioned in previous metanalysis about the possible use of habitat suitability models as proxy of abundance. In the revised version, we add a few lines (see lines 370-374; line 441 and lines 463-469) in this regard to enrich the discussion and provide insights into this hotly debated relationship (see metanalysis of Weber et al 2017).
4) Surveys were conducted between 1 April and 31 August. Since April is the main spring migration period of Montagu’s harrier many migrants may pass the area, which may utilize different habitats than breeding birds. Combining both types of birds – breeding and migrating individuals – may affect the predicted habitat suitability. This is an issue, which should be also handled somehow, e.g. by excluding data that might consider migratory individuals (e.g. by excluding data collected outside the main breeding period [April?; August?]. You stated that Cir pyg is migratory, but I will guess that your data might also contain migratory Cir cya individuals going on their northern breeding areas?
Response: Sorry, it was my mistake, we revised the starting date of the fieldwork and the surveys started from the second half of April. Anyway, regarding the migratory behavior of the Montagu’s harrier and the dispersive character of the Hen harrier, these populations are very philopatric, that is, individuals tend to return to breed in this area every year (sometimes even in the same shrub patch). In addition, the fledglings stay in the breeding area until September (we did not observe any flocks of individuals assembling before migration, i.e., no migratory restlessness in August). We now add a few lines in the survey design section to correct the dates of both surveys, between 15th Abril and 31st August (in 1997 and 2017), and explain the philopatric behavior of these species, which strongly reduce the risk of recording individuals going northern areas (lines 156 and 160).
5) You car-based point-count (very shorht survey time per survey) methods is not very common, and seldomly use. I am not sure how suitable and efficient is it to conduct survey for (raptor) species having large territories. Do you have any references to support the use of this method?
Response: Here we quote a paragraph from the manuals available at the Raptor Research Foundation, page 93 in the chapter of Anderson (https://www.raptorresearchfoundation.org/publications/techniques-manual/): “Surveys for raptors often have been conducted along roads where raptors are observed and counted from vehicles (e.g., Andersen et al. 1985). Surveys along roads have been used to describe raptor distribution (e.g., Yosef et al. 1999, Bak et al. 2001), diversity (e.g., Ross et al. 2003), relative abundance in relation to land-use practices (e.g., Sorley and Andersen 1994, Yahner and Rohrbaugh 1998, Williams et al. 2000), and habitat use at broad spatial scales (e.g., Garner and Bednarz 2000, Olson and Arsenault 2000). Studies of raptor behavior (e.g., Manosa et al. 1998, Rejt 2001), food habits (e.g., Dekker 1995, Kaltenecker et al. 1998) or population dynamics (e.g., Kerlinger and Lein 1988, Hiraldo et al. 1995, Bridgeford and Bridgeford 2003) also have been based on surveys along roads. Surveys from roads also have been used to locate nests in natural (e.g., Travaini et al. 1994, Woodbridge et al. 1995)”.
To support our survey methods, we now encourage readers to see a key reference for raptor surveys and reference therein (to avoid include many citations; lines 162-163).
Andersen, D. E. Survey techniques. In Raptor Research and Management Techniques Manual; Bird, D. M.; Bildstein, K. L., Eds.; 2007; pp. 89–100.
Below some my own comments:
-use capital letters for the common bird names; Hen Harrie; Montagu’s Harrier
Response: Done.
-take statistics away from the abstract
Response: Done.
-L34: "AUC and TSS"; open acronyms
Response: Done.
-L51: "Unfortunately, long-term time series of population abundance (obtained from 51 standardized monitoring protocols involving repeated surveys of the same population) 52 are rarely available"; I do not agree that, e.g. in Finland, U.K., USA, there are a long history of bird monitoring. Even from raptor species in Finland, see e.g. E. Korpimäki´s extesive work with Kestrel and owls.
Response: we now state that: “Long-term time series of population abundance (obtained from standardized monitoring protocols involving repeated surveys of the same population) are essential for their management but very costly and time-consuming” (lines 59-60) to explain why model-assisted monitoring has recently emerged as an alternative, cost-effective way of assessing the impact of environmental change on biodiversity.
-prey availability (cyclicity) may influence your results, e.g. voles are important food for both Harrie species, how this is taken into account in your stydy, or do you think this is not an important factor in you case study; you should indicate this e.g. in the method section descriping the study species or in the discussion. Indeed, you have only two study years, what about the prey availability during these two years?
Response: We agree that this is an important issue/limitation that cannot be addressed in diachronic studies (i.e., two years or snapshots) but it requires long-term sampling protocols or remote sensing time series. We now include in the discussion and conclusions a few lines to remark the importance of interannual variation in prey availability and how remotely sensed functional variables have been recently used as proxy of interannual fluctuations in food resources (see e.g., Requena-Mullor et al., 2014; Regos et al., 2021, among others) (see lines 374-378, and 463-469).
Requena-Mullor, J.M., López, E., Castro, A.J., Cabello, J., Virgós, E., González-Miras, E., Castro, H., 2014. Modeling spatial distribution of European badger in arid landscapes: an ecosystem functioning approach. Landsc. Ecol. 29, 843–855. https:// doi.org/10.1007/s10980-014-0020-4.
Regos, A., Arenas-Castro, S., Tapia, L., Domínguez, J., & Honrado, J. P. (2021). Using remotely sensed indicators of primary productivity to improve prioritization of conservation areas for top predators. Ecological Indicators, 125, 107503.
-L90: indicate, what are the European and national treath status of the study species
Response: We now specify that both species are included in the Annex I of the European Bird Directive with conservation status ‘Vulnerable’ at national level (lines 108-109; see Arroyo et al 2019).
Arroyo, B., Molina, B. y Del Moral, J. C. 2019. El aguilucho cenizo y el aguilucho pálido en España. Población reproductora en 2017 y método de censo. SEO/BirdLife. Madrid.
-L145-145: add test statistics
Response: Here we just indicate to the total number of individuals recorded per year.
-Fig. 1: add the scale for the low-right part local figure; I noted that one black and open dot at the midlle are located very near of each other; independent survey stations?
Response: we now add a scale on the low-right part of the figure. All points along the transects were established at intervals of 1 km to avoid pseudoreplication, but dots in the figure are large to facilitate their visualization. Pseudoreplication was very carefully considered during the fieldwork.
-Table 1: give SD and do statistical testing and give test values
Response: Table 1 provides the extent (in hectares) covered by each land-cover/use type in our study area per year, so not sure what kind of dispersal metrics or statistics should be carried out.
-Figure 2: Correct Y-axis title writing "Variableimportance"
Response: done.
-figure 5: I will gues that the statistics are rS (with S a subscript)
Response: We now clearly refer to the Spearman's ρ and Pearson’s r.
-Discussion: describe clearly the weaknesses of your study.
Response: In the revised version, we have described the weaknesses of our study associated with the autoecology of our target species (e.g., the semicolonial and migratory behavior of the Montagu’s harrier) and the interannual variation in prey availability that has not been mentioned in previous metanalysis (Weber et al. 2017) to provide new insights into this hotly debated topic of using ecological niche models as proxy for abundance (lines 370 and 389; and 463-469).
Weber, M. M.; Stevens, R. D.; Diniz-Filho, J. A. F.; Grelle, C. E. V. Is there a correlation between abundance and environmental suitability derived from ecological niche modelling? A meta-analysis. Ecography (Cop.). 2017, 40, 817–828.
-Conclusions: suggest some future study ideas; e.g. using marked individuals etc.
Response: This is indeed a good point. We have now added a few future ideas to overcome the inherent difficulties associated with the migratory and semicolonial behavior of the Montagu’s harrier. We highlighted the need of sampling protocols supported by marked individuals that allow tracking their habitat occupancy along the annual cycle or improving ENMs by including time series of remotely sensed variables as proxy for habitat dynamics (e.g., interannual variations in prey availability) (lines 463-469).
-"Data Availability Statement: In this section, please provide details regarding where data supporting reported results can be found, including links to publicly archived datasets analyzed or generated during the study. Please refer to suggested Data Availability Statements in section “MDPI Research Data Policies” at https://www.mdpi.com/ethics. You might choose to exclude this statement if the study did not report any data." EDIT THIS SECTION
Response: Landsat images are freely available from the United States Geological Survey (USGS) Global Visualization Viewer (http://glovis.usgs.gov). Topographic information was obtained from NASA’s Shuttle Radar Topography Mission (GDEM-SRTM) (https://www2.jpl.nasa.gov/srtm/). Raptor data are available from the corresponding author on reasonable request.
Please, respond all the comments point-by-point when you will return the revised version of your work.
Sincerely
Jukka Jokimäki
Section Editor-in-Chief
Birds, Animals
Reviewer#1
The manuscript is consistently very well written, both in terms of language and in terms of content!
Response: Thank you very much for your time and positive feedback. We strongly appreciate your comments and suggestions that helped us to improve our work in key issues that ecological modelers must take into account when using habitat suitability predicted by ENMs as proxy for abundance.
However, that the temporal variation in species abundance could not be significantly explained by changes in habitat suitability predicted by the ENMs is indeed remarkable, when considering the dramatic change of open and closed heathland cover, the two variables which proved being extremely important predictors in the calculated ENMs. Hence, I have some doubts as to whether the data is of sufficient quality to reliably examine the questions addressed, for the following reasons:
(1) The study area is pretty small (only 33.5 km2) for a study on raptors. Can the authors provide an estimate on the number of breeding pairs of both species, which bred in the area in individual years? When most of the data points are generated by only few breeding pairs, pseudoreplication may be a substantial problem!
Response: We estimated 4 breeding pairs for Hen harrier in 1997 that decreased up to 2 in 2017; and 3-4 pairs of Montagu’s harries in both years. The observers (which are also co-authors of the work) has a vast experience (more than 20 years) working with raptors in our region. We are aware of the implications of pseudoreplication, so this issue has been carefully considered during the sampling protocols. Within each transect every individual was controlled to avoid doble counting (now clarified in line 167). The transects were carried out several times over the breeding period to ensure that all absences are ‘real’ and not ‘pseudo-absences’ due to an insufficient sampling effort. Within each transect every individual was controlled to avoid pseudo-replication. If several observations are recorded in one sampling point across the breeding period, the record remains as “presence”. On the contrary, if after all surveys there were no observations, it would be recorded as “absence”. We think that our sampling protocol allows to be confidence about the habitat characterization and preference of this population in our study area. However, following the Editor’s and reviewer’s suggestions, we now include a few lines in the discussion regarding potential problems associated with pseudo-replication (see lines 370 and 373).
Regarding the study area, it is small but very representative of the European Atlantic mountain heathlands that these species has historically used to breed in NW Spain. Unfortunately, mountain heathlands are nowadays under threat and very restricted in southern Europe, being priority conservation areas in Spain (Rego et al 2013; Fagúndez et al 2013). This area is included in a Natura 2000 site, being one of the core breeding areas of these two species in our region. We have now remarked in the revised version the representativeness of our study area in the European context to help the audience (including researchers working with these species) understand the relevance of our results for other Atlantic Mountain heathlands (lines 140 and 144-145); and two new references (highlighted in yellow):
Rego, P. R.; Guitián, M. A. R.; Castro, H. L.; Da Costa, J. F.; Sobrino, C. M. Loss of european dry heaths in NW Spain: A case study. Diversity 2013, 5, 557–580.
Fagúndez, J. Heathlands confronting global change: Drivers of biodiversity loss from past to future scenarios. Ann. Bot. 2013, 111, 151–172.
In addition, we think that the fine-scale habitat characterization based on Landsat images (30-m resolution) allows to model these species with confidence. In fact, land-use changes and their impacts on habitat quality predicted by our models are in line with other studies carried out in our region at larger scale. See our previous studies:
Regos, A.; Tapia, L.; Gil-Carrera, A.; Domínguez, J. Monitoring protected areas from space: A multi-temporal assessment using raptors as biodiversity surrogates. PLoS One 2017, 12, e0181769.
Tapia, L.; Dominguez, J.; Rodriguez, L. Modelling habitat selection and distribution of Hen harrier (Circus cyaneus) and Montagu’s harrier (Circus pygargus) in a mountainous area in Galicia. J. Raptor Res. 2004, 38, 133–140. 441
Tapia, L.; Domínguez, J.; Rodríguez, L. Hunting habitat preferences of raptors in a mountainous area (northwestern Spain). Polish J. Ecol. 2008, 56, 323–333.
Tapia, L.; Regos, A.; Gil-Carrera, A.; Dominguez, J. Assessing the temporal transferability of raptor distribution models: Implications for conservation. Bird Conserv. Int. 2018, 28, 375–389. 450
Tapia, L.; Regos, A.; Gil-Carrera, A.; Domínguez, J. Unravelling the response of diurnal raptors to land use change in a highly dynamic landscape in northwestern Spain: an approach based on satellite earth observation data. Eur. J. Wildl. Res. 2017, 63, 40.
(2) A further emerging problem may be that although breeding Hen Harriers predominately hunt within 1-2 km from their nest, they can travel up to 9 km from their nests and can have an average home-range of 8 km2 (Arroyo et al. 2014, Bird Study 61/1). Hence, it is very likely that the same birds were counted at various census points. Again, then pseudoreplication / spatial autocorrelation may severely weaken the reliability of the models.
Response: The observers (who are also co-authors of the work) have a very long experience working with raptors and with these species in our study area. They were aware of the risk of pseudoreplication but, as commented above, every contact within a transect was attribute to one individual, the risk of the same individual was counted twice within the same transect was very low. The replication of the transect over time ensure that all absences recorded are real and not pseudo-absences (critical issue in habitat suitability modelling). The number of individuals recorded in each transect is low, which make them easily controllable during the survey. We are therefore confident that pseudo-replication and spatial autocorrelation was not a problem, as that risk was carefully considered by observers when sampling the populations. However, we have included in the discussion a few lines to make the readers aware of the risk associated with these issues in the case of species with large home ranges such as these raptors, in particular the large hunting distances of breeding male and female Hen Harriers (lines 370-374). We have also included the reference suggested by the reviewer to support these aspects/limitations.
(3) In Montagu’s harrier the situation may be even more problematic since this species is semi-colonial (e.g. Arroyo et al. 2004, BWP Update 6: 41-55). This may explain why predictions of habitat suitability based on occurrence data do not result in close correlations with spatio-temporal abundance changes.
Response: thanks very much for this pertinent comment and the suggested reference (now included). The semi-colonial behavior of the Montagu’s harrier can be another factor that could explain why predictions of habitat suitability based on occurrence are not strongly correlated with spatiotemporal changes in abundance. This is in fact an issue that has not been mentioned in previous metanalysis about the use of habitat suitability models as proxy for abundance, and autoecology of our target species can provide insights into this hotly debated relationship (see metanalysis of Weber et al. 2017). We add a few lines in this regard to enrich the discussion and contribute to the debate (lines 370 and 374) together with recommendations to overcome these limitations (lines 463-469).
At least it has to be comprehensively addressed in the discussion to what extent these two issues (2) and (3) may affect the performance of the calculated models!
Response: In the revised version, we have now explicitly addressed these issues particularly important for migratory, semi-colonial species with large home range. We think that this cannot only improve our work but also call the attention to important issues to be considered in future studies when using habitat suitability models to infer spatiotemporal changes in abundance, especially with raptors (lines 370 and 374; line 441, and lines 463-469).
(4) Surveys were conducted between 1 April and 31 August. Since April is the main spring migration period of Montagu’s harrier many migrants may pass the area, which may utilize different habitats than breeding birds. Combining both types of birds – breeding and migrating individuals – may affect the predicted habitat suitability. This is an issue, which should be also clarified.
Response: Sorry, it was my mistake, my colleagues confirmed that the starting date of the fieldwork and the surveys started from the second half of April. Anyway, regarding the migratory behavior of the Montagu’s harrier and the dispersive character of the Hen harrier, these populations are very philopatric, that is, individuals tend to return to breed in this area every year (sometimes even in the same shrub patch). In addition, the fledglings stay in the breeding area until September (we did not observe any flocks of individuals assembling before migration, i.e., no migratory restlessness in August). We now add a few lines in the survey design section to correct the dates of both surveys, between 15th Abril and 31st August (in 1997 and 2017), and explain the philopatric behavior of these species, which strongly reduce the risk of recording individuals going northern areas (lines 156 and 160).
Specific comments
It is unnecessary in the text of the discussion to refer to specific figures again in parentheses.
Response: We removed any reference to specific figures in the discussion.
Figure 2: I fear that in this black-and-white figure not everything can be seen that is necessary for the interpretation of the results. Please check.
Response: All figures are in color. Figure 2 should show the relative importance of each predictor variable. We have just realized that figures 2, 4 and 5 are wrong in the PDF (not in the Word file). It might be caused by the automatic conversion from Word to PDF during the submission process. We apologies for this error. We will ask the editorial team for the possibility of uploading the figures as separate files to ensure a correct visualization.
Here all original figures:
https://nubeusc-my.sharepoint.com/:f:/g/personal/adrian_regos_usc_es/Eg2jXBBLDMROt4kOA8BvzxYB7xq6QwzQGQ2EOiYvt9SNSg?e=A0hKxG
Editorial team: Please ensure that the revised version include the figures with colors. Thank you!
Reviewer#2
The paper entitled “Caution is needed when using niche models to infer changes in species abundance: the case of two sympatric raptor populations” submitted to Animals deals with habitat loss affecting populations of two sympatric harrier species in Spain.
The MS is well-written and I only have some minor revisions before acceptance.
Response: thank you very much for your time and positive feedback on our work. We have addressed all your comments to improve our work.
- Lines 46-47. I guess it is the most important cause of biodiversity loss at all.
Response: Yes, agree. There are several global assessments that point out to habitat loss as the most important threat (see e.g., Diaz et al 2019). We now state that habitat loss is the most important cause of biodiversity loss (line 54).
Reference:
Díaz, S.; et al. Summary for policymakers of the global assessment report on biodiversity and ecosystem services of the Intergovernmental Science-Policy Platform 409 on Biodiversity and Ecosystem Services; IPBES secretariat: Bonn, Germany, 2019.
- Line 57. Not only environmental covariates, also climatic ones!
Response: we now clarify that the environmental covariates are mostly represented by climatic and topographic variables, being widely used in niche modelling due to their large availability despite the relevance of local habitat variables (lines 64-65).
- Lines 100-110. It is not exactly clear to me why authors selected exactly those species and why it is so important to study these harrier species. Authors say that they are affected by habitat loss, but how? At lines 97-99, you talk about decline. How much are they declining and in which time?
Response: Previous studies developed by our team showed important declines for these open-habitat raptors in the province, where the study area is located, over the last 20 years. These changes in habitat availability and quality were caused by land-use conversions from heathlands and grasslands to fast-growing tree plantations (mostly Pinus but also Eucalyptus spp.) and native forest expansion due to rural abandonment processes (loss of traditional agricultural and livestock practices) and intensive forest practices. We now added this complementary information to help readers understand the suitability of these two species as case study, in addition to their relevance in terms of conservation of mountain heathlands and other species linked to these habitats (see lines 113-118).
References:
Regos, A.; Tapia, L.; Gil-Carrera, A.; Domínguez, J. Monitoring protected areas from space: A multi-temporal assessment using raptors as biodiversity surrogates. PLoS One 2017, 12, e0181769.
Tapia, L.; Dominguez, J.; Rodriguez, L. Modelling habitat selection and distribution of Hen harrier (Circus cyaneus) and Montagu’s harrier (Circus pygargus) in a mountainous area in Galicia. J. Raptor Res. 2004, 38, 133–140. 441
Tapia, L.; Domínguez, J.; Rodríguez, L. Hunting habitat preferences of raptors in a mountainous area (northwestern Spain). Polish 442 J. Ecol. 2008, 56, 323–333.
Tapia, L.; Regos, A.; Gil-Carrera, A.; Dominguez, J. Assessing the temporal transferability of raptor distribution models: Implications for conservation. Bird Conserv. Int. 2018, 28, 375–389. 450
Tapia, L.; Regos, A.; Gil-Carrera, A.; Domínguez, J. Unravelling the response of diurnal raptors to land use change in a highly dynamic landscape in northwestern Spain: an approach based on satellite earth observation data. Eur. J. Wildl. Res. 2017, 63, 40.
Regos, A., Arenas-Castro, S., Tapia, L., Domínguez, J., & Honrado, J. P. (2021). Using remotely sensed indicators of primary productivity to improve prioritization of conservation areas for top predators. Ecological Indicators, 125, 107503.
- Lines 105-110. Authors should include some predictions.
Response: We agree with the reviewer that the last part of the introduction also benefit from including some predictions or expectations regarding our hypotheses and analysis. However, our predictions were already stated in lines 93-97: “we would expect model-predicted environmental suitability to be significantly correlated with species abundance, thus supporting the use of SRS-based ENMs to estimate species abundance. If so, both the spatial and temporal variation in species abundance should be explained (at least partly) by changes in environmental suitability predicted by these models”.
- Methods are clearly stated, and replicable.
Response: Thank you!
- I totally do not understand Figure 2, it is almost unreadable and provide me with no information.
Response: Figure 2 should show the relative importance of each predictor variable. We have just realized that figures 2, 4 and 5 were wrong in the PDF (not in the Word file). It might be caused by the automatic conversion from Word to PDF during the submission process. We apologize for this error. We will ask editorial team for the possibility of uploading the figures as separate files to ensure a correct visualization.
Anyway, here you can find all original figures:
https://nubeusc-my.sharepoint.com/:f:/g/personal/adrian_regos_usc_es/Eg2jXBBLDMROt4kOA8BvzxYB7xq6QwzQGQ2EOiYvt9SNSg?e=A0hKxG
- Line 289. Delete “study” before “findings”.
Response: done.
- Line 295. You say “various authors” but you cited only one work, please increase literature search or change “various authors” with “one study”.
Response: We only cited one work because that study is a metanalysis about that topic. We now encourage readers to see “reference therein” and include another key reference.
Line 321. Delete “the” before “findings”.
Response: done.

Reviewer 2 Report
The paper entitled “Caution is needed when using niche models to infer changes in species abundance: the case of two sympatric raptor populations” submitted to Animals deals with habitat loss affecting populations of two sympatric harrier species in Spain.
The MS is well-written and I only have some minor revisions before acceptance.
- Lines 46-47. I guess it is the most important cause of biodiversity loss at all.
- Line 57. Not only environmental covariates, also climatic ones!
- Lines 100-110. It is not exactly clear to me why authors selected exactly those species and why it is so important to study these harrier species. Authors say that they are affected by habitat loss, but how? At lines 97-99, you talk about decline. How much are they declining and in which time?
- Lines 105-110. Authors should include some predictions.
- Methods are clearly stated, and replicable.
- I totally do not understand Figure 2, it is almost unreadable and provide me with no information.
- Line 289. Delete “study” before “findings”.
- Line 295. You say “various authors” but you cited only one work, please increase literature search or change “various authors” with “one study”.
- Line 321. Delete “the” before “findings”.
Academic Editor Notes
I have now received reports from the two reviewers for your ms, and I have read the ms by myself. One reviewer has given only minor comments, whereas the other (R1) has pointed some major concerns related on your method/data quality. I agree with the R2. Therefore, your ms will need a major revision before I can consider it for the publication for the Animals.
Dear Editor, Jukka Jokimäki,
We are pleased to submit a revised version of the manuscript ‘1265920’, entitled “Caution is needed when using niche models to infer changes in species abundance: the case of two sympatric raptor populations”. First, we would like to thank you for handling our manuscript and your personal and careful revision. We are also grateful to the two reviewers, whose comments and suggestions helped us to improve the quality of our manuscript.
In this revised version, we addressed thoroughly all comments made by the Editor and reviewers. Specifically, regarding the reviewers’ major concerns, we have better justified the suitability of the survey methods and discussed potential limitations related to the migratory and semi-colonial behavior of the Montagu’s harrier. Still, we have also explained that pseudoreplication has been considered during the surveys and why is not an issue of special concern in our case. We have also discussed other limitations such as the interannual variation in prey availability and how this could be addressed in future studies. The remaining minor issues have also been addressed and/or clarified in the revised manuscript.
In ‘response to reviewers' comments’, we provide our point-by-point responses to all comments or suggestions raised by the reviewers, indicating the lines where changes have been included in the revised version of the manuscript. The clean version of the revised manuscript is tagged as ‘Clean Version’. Changes are highlighted by using the ‘tracked changes mode’ in MS Word, after the clean version. In the detailed point-by-point responses we refer to the line numbers of the “clean” version of the manuscript for an easier reading.
Please note that Figures 2, 4 and 5 were wrong in the PDF of the original submission (not in the Word file). It might have been caused by the automatic conversion from Word to PDF during the submission process. We apologize for this error. We want to ask the Editorial team for the possibility of uploading the figures as separate files to ensure a correct visualization.
The figures (in pdf and tiff) are in the zip file but also in this personal link:
https://nubeusc-my.sharepoint.com/:f:/g/personal/adrian_regos_usc_es/Eg2jXBBLDMROt4kOA8BvzxYB7xq6QwzQGQ2EOiYvt9SNSg?e=A0hKxG
We hope that this revised version of the manuscript meets the journal’s expectations for publication.
We look forward for your editorial opinion.
Sincerely,
Adrián Regos on behalf of all co-authors.
----------------------------
The major concerns are related to:
1) The study area is pretty small (only 33.5 km2) for a study on raptors
Response: The study area is small but very representative of the European Atlantic mountain heathlands that these species has historically used to breed in NW Spain. Unfortunately, mountain heathlands are nowadays under threat and very restricted in southern Europe, being priority conservation areas in Spain (Regos et al 2013; Fagúndez et al 2013). This area is included in a Natura 2000 site, being one of the core breeding areas of these two species in our region. We have now remarked in the revised version the representativeness of our study area in the European context to help the audience (including researchers working with these species) understand the relevance of our results for other Atlantic Mountain heathlands (lines 140 and 144-145); and two new references (highlighted in yellow):
Rego, P. R.; Guitián, M. A. R.; Castro, H. L.; Da Costa, J. F.; Sobrino, C. M. Loss of european dry heaths in NW Spain: A case study. Diversity 2013, 5, 557–580.
Fagúndez, J. Heathlands confronting global change: Drivers of biodiversity loss from past to future scenarios. Ann. Bot. 2013, 111, 151–172.
In addition, we think that the fine-scale habitat characterization based on Landsat images (30-m resolution) allows to model these species with confidence. In fact, land-use changes and their impacts on habitat quality predicted by our models are in line with other studies carried out in our region at larger scale. See our previous studies:
Regos, A.; Tapia, L.; Gil-Carrera, A.; Domínguez, J. Monitoring protected areas from space: A multi-temporal assessment using raptors as biodiversity surrogates. PLoS One 2017, 12, e0181769.
Tapia, L.; Dominguez, J.; Rodriguez, L. Modelling habitat selection and distribution of Hen harrier (Circus cyaneus) and Montagu’s harrier (Circus pygargus) in a mountainous area in Galicia. J. Raptor Res. 2004, 38, 133–140. 441
Tapia, L.; Domínguez, J.; Rodríguez, L. Hunting habitat preferences of raptors in a mountainous area (northwestern Spain). Polish J. Ecol. 2008, 56, 323–333.
Tapia, L.; Regos, A.; Gil-Carrera, A.; Dominguez, J. Assessing the temporal transferability of raptor distribution models: Implications for conservation. Bird Conserv. Int. 2018, 28, 375–389. 450
Tapia, L.; Regos, A.; Gil-Carrera, A.; Domínguez, J. Unravelling the response of diurnal raptors to land use change in a highly dynamic landscape in northwestern Spain: an approach based on satellite earth observation data. Eur. J. Wildl. Res. 2017, 63, 40.
2) There is a possibility of the pseudoreplication
Response: the observers (which are also co-authors of the work) has a vast experience (more than 20 years) working with raptors in our region. We are aware of the implications of pseudoreplication, so this issue has been carefully considered during the sampling protocols. Within each transect every individual was controlled to avoid doble counting. The advantage of our protocol is that every transect was carried out several times over the breeding period to ensure that all absences are “real” and not “pseudo-absence” due to an insufficient sampling effort. If several observations are recorded in one sampling point across the breeding period, the record remains as “presence”. On the contrary, if after all surveys there were no observations, it would be recorded as “absence”. We think that our sampling protocol allows to be confidence about the habitat characterization and preference of this population in our study area. However, following the Editor’s and reviewer#1 suggestions, we now discuss potential problems associated with pseudo-replication (see lines 370 and 373).
3) One of your study species, Montagu’s harrier, is semi-colonial
At least it has to be comprehensively addressed in the discussion to what extent these three issues have been tried to controlled during the field methods, and how these topics may affect the performance of the calculated models!
Response: We agree that this issue deserves to be mentioned for the case of Montagu’s harrier for readers that are not familiar with these species. The observers have a deep knowledge of the ecology of both species. The semi-colonial behavior of the Montagu’s harrier was considered during the transect by controlling all individuals recorded at each point. The probability of double counting the same individual during the transect was minimum (now clarified in line 167). However, the semicolonial behavior of some species is in fact a very interesting issue that has not been mentioned in previous metanalysis about the possible use of habitat suitability models as proxy of abundance. In the revised version, we add a few lines (see lines 370-374; line 441 and lines 463-469) in this regard to enrich the discussion and provide insights into this hotly debated relationship (see metanalysis of Weber et al 2017).
4) Surveys were conducted between 1 April and 31 August. Since April is the main spring migration period of Montagu’s harrier many migrants may pass the area, which may utilize different habitats than breeding birds. Combining both types of birds – breeding and migrating individuals – may affect the predicted habitat suitability. This is an issue, which should be also handled somehow, e.g. by excluding data that might consider migratory individuals (e.g. by excluding data collected outside the main breeding period [April?; August?]. You stated that Cir pyg is migratory, but I will guess that your data might also contain migratory Cir cya individuals going on their northern breeding areas?
Response: Sorry, it was my mistake, we revised the starting date of the fieldwork and the surveys started from the second half of April. Anyway, regarding the migratory behavior of the Montagu’s harrier and the dispersive character of the Hen harrier, these populations are very philopatric, that is, individuals tend to return to breed in this area every year (sometimes even in the same shrub patch). In addition, the fledglings stay in the breeding area until September (we did not observe any flocks of individuals assembling before migration, i.e., no migratory restlessness in August). We now add a few lines in the survey design section to correct the dates of both surveys, between 15th Abril and 31st August (in 1997 and 2017), and explain the philopatric behavior of these species, which strongly reduce the risk of recording individuals going northern areas (lines 156 and 160).
5) You car-based point-count (very shorht survey time per survey) methods is not very common, and seldomly use. I am not sure how suitable and efficient is it to conduct survey for (raptor) species having large territories. Do you have any references to support the use of this method?
Response: Here we quote a paragraph from the manuals available at the Raptor Research Foundation, page 93 in the chapter of Anderson (https://www.raptorresearchfoundation.org/publications/techniques-manual/): “Surveys for raptors often have been conducted along roads where raptors are observed and counted from vehicles (e.g., Andersen et al. 1985). Surveys along roads have been used to describe raptor distribution (e.g., Yosef et al. 1999, Bak et al. 2001), diversity (e.g., Ross et al. 2003), relative abundance in relation to land-use practices (e.g., Sorley and Andersen 1994, Yahner and Rohrbaugh 1998, Williams et al. 2000), and habitat use at broad spatial scales (e.g., Garner and Bednarz 2000, Olson and Arsenault 2000). Studies of raptor behavior (e.g., Manosa et al. 1998, Rejt 2001), food habits (e.g., Dekker 1995, Kaltenecker et al. 1998) or population dynamics (e.g., Kerlinger and Lein 1988, Hiraldo et al. 1995, Bridgeford and Bridgeford 2003) also have been based on surveys along roads. Surveys from roads also have been used to locate nests in natural (e.g., Travaini et al. 1994, Woodbridge et al. 1995)”.
To support our survey methods, we now encourage readers to see a key reference for raptor surveys and reference therein (to avoid include many citations; lines 162-163).
Andersen, D. E. Survey techniques. In Raptor Research and Management Techniques Manual; Bird, D. M.; Bildstein, K. L., Eds.; 2007; pp. 89–100.
Below some my own comments:
-use capital letters for the common bird names; Hen Harrie; Montagu’s Harrier
Response: Done.
-take statistics away from the abstract
Response: Done.
-L34: "AUC and TSS"; open acronyms
Response: Done.
-L51: "Unfortunately, long-term time series of population abundance (obtained from 51 standardized monitoring protocols involving repeated surveys of the same population) 52 are rarely available"; I do not agree that, e.g. in Finland, U.K., USA, there are a long history of bird monitoring. Even from raptor species in Finland, see e.g. E. Korpimäki´s extesive work with Kestrel and owls.
Response: we now state that: “Long-term time series of population abundance (obtained from standardized monitoring protocols involving repeated surveys of the same population) are essential for their management but very costly and time-consuming” (lines 59-60) to explain why model-assisted monitoring has recently emerged as an alternative, cost-effective way of assessing the impact of environmental change on biodiversity.
-prey availability (cyclicity) may influence your results, e.g. voles are important food for both Harrie species, how this is taken into account in your stydy, or do you think this is not an important factor in you case study; you should indicate this e.g. in the method section descriping the study species or in the discussion. Indeed, you have only two study years, what about the prey availability during these two years?
Response: We agree that this is an important issue/limitation that cannot be addressed in diachronic studies (i.e., two years or snapshots) but it requires long-term sampling protocols or remote sensing time series. We now include in the discussion and conclusions a few lines to remark the importance of interannual variation in prey availability and how remotely sensed functional variables have been recently used as proxy of interannual fluctuations in food resources (see e.g., Requena-Mullor et al., 2014; Regos et al., 2021, among others) (see lines 374-378, and 463-469).
Requena-Mullor, J.M., López, E., Castro, A.J., Cabello, J., Virgós, E., González-Miras, E., Castro, H., 2014. Modeling spatial distribution of European badger in arid landscapes: an ecosystem functioning approach. Landsc. Ecol. 29, 843–855. https:// doi.org/10.1007/s10980-014-0020-4.
Regos, A., Arenas-Castro, S., Tapia, L., Domínguez, J., & Honrado, J. P. (2021). Using remotely sensed indicators of primary productivity to improve prioritization of conservation areas for top predators. Ecological Indicators, 125, 107503.
-L90: indicate, what are the European and national treath status of the study species
Response: We now specify that both species are included in the Annex I of the European Bird Directive with conservation status ‘Vulnerable’ at national level (lines 108-109; see Arroyo et al 2019).
Arroyo, B., Molina, B. y Del Moral, J. C. 2019. El aguilucho cenizo y el aguilucho pálido en España. Población reproductora en 2017 y método de censo. SEO/BirdLife. Madrid.
-L145-145: add test statistics
Response: Here we just indicate to the total number of individuals recorded per year.
-Fig. 1: add the scale for the low-right part local figure; I noted that one black and open dot at the midlle are located very near of each other; independent survey stations?
Response: we now add a scale on the low-right part of the figure. All points along the transects were established at intervals of 1 km to avoid pseudoreplication, but dots in the figure are large to facilitate their visualization. Pseudoreplication was very carefully considered during the fieldwork.
-Table 1: give SD and do statistical testing and give test values
Response: Table 1 provides the extent (in hectares) covered by each land-cover/use type in our study area per year, so not sure what kind of dispersal metrics or statistics should be carried out.
-Figure 2: Correct Y-axis title writing "Variableimportance"
Response: done.
-figure 5: I will gues that the statistics are rS (with S a subscript)
Response: We now clearly refer to the Spearman's ρ and Pearson’s r.
-Discussion: describe clearly the weaknesses of your study.
Response: In the revised version, we have described the weaknesses of our study associated with the autoecology of our target species (e.g., the semicolonial and migratory behavior of the Montagu’s harrier) and the interannual variation in prey availability that has not been mentioned in previous metanalysis (Weber et al. 2017) to provide new insights into this hotly debated topic of using ecological niche models as proxy for abundance (lines 370 and 389; and 463-469).
Weber, M. M.; Stevens, R. D.; Diniz-Filho, J. A. F.; Grelle, C. E. V. Is there a correlation between abundance and environmental suitability derived from ecological niche modelling? A meta-analysis. Ecography (Cop.). 2017, 40, 817–828.
-Conclusions: suggest some future study ideas; e.g. using marked individuals etc.
Response: This is indeed a good point. We have now added a few future ideas to overcome the inherent difficulties associated with the migratory and semicolonial behavior of the Montagu’s harrier. We highlighted the need of sampling protocols supported by marked individuals that allow tracking their habitat occupancy along the annual cycle or improving ENMs by including time series of remotely sensed variables as proxy for habitat dynamics (e.g., interannual variations in prey availability) (lines 463-469).
-"Data Availability Statement: In this section, please provide details regarding where data supporting reported results can be found, including links to publicly archived datasets analyzed or generated during the study. Please refer to suggested Data Availability Statements in section “MDPI Research Data Policies” at https://www.mdpi.com/ethics. You might choose to exclude this statement if the study did not report any data." EDIT THIS SECTION
Response: Landsat images are freely available from the United States Geological Survey (USGS) Global Visualization Viewer (http://glovis.usgs.gov). Topographic information was obtained from NASA’s Shuttle Radar Topography Mission (GDEM-SRTM) (https://www2.jpl.nasa.gov/srtm/). Raptor data are available from the corresponding author on reasonable request.
Please, respond all the comments point-by-point when you will return the revised version of your work.
Sincerely
Jukka Jokimäki
Section Editor-in-Chief
Birds, Animals
Reviewer#1
The manuscript is consistently very well written, both in terms of language and in terms of content!
Response: Thank you very much for your time and positive feedback. We strongly appreciate your comments and suggestions that helped us to improve our work in key issues that ecological modelers must take into account when using habitat suitability predicted by ENMs as proxy for abundance.
However, that the temporal variation in species abundance could not be significantly explained by changes in habitat suitability predicted by the ENMs is indeed remarkable, when considering the dramatic change of open and closed heathland cover, the two variables which proved being extremely important predictors in the calculated ENMs. Hence, I have some doubts as to whether the data is of sufficient quality to reliably examine the questions addressed, for the following reasons:
(1) The study area is pretty small (only 33.5 km2) for a study on raptors. Can the authors provide an estimate on the number of breeding pairs of both species, which bred in the area in individual years? When most of the data points are generated by only few breeding pairs, pseudoreplication may be a substantial problem!
Response: We estimated 4 breeding pairs for Hen harrier in 1997 that decreased up to 2 in 2017; and 3-4 pairs of Montagu’s harries in both years. The observers (which are also co-authors of the work) has a vast experience (more than 20 years) working with raptors in our region. We are aware of the implications of pseudoreplication, so this issue has been carefully considered during the sampling protocols. Within each transect every individual was controlled to avoid doble counting (now clarified in line 167). The transects were carried out several times over the breeding period to ensure that all absences are ‘real’ and not ‘pseudo-absences’ due to an insufficient sampling effort. Within each transect every individual was controlled to avoid pseudo-replication. If several observations are recorded in one sampling point across the breeding period, the record remains as “presence”. On the contrary, if after all surveys there were no observations, it would be recorded as “absence”. We think that our sampling protocol allows to be confidence about the habitat characterization and preference of this population in our study area. However, following the Editor’s and reviewer’s suggestions, we now include a few lines in the discussion regarding potential problems associated with pseudo-replication (see lines 370 and 373).
Regarding the study area, it is small but very representative of the European Atlantic mountain heathlands that these species has historically used to breed in NW Spain. Unfortunately, mountain heathlands are nowadays under threat and very restricted in southern Europe, being priority conservation areas in Spain (Rego et al 2013; Fagúndez et al 2013). This area is included in a Natura 2000 site, being one of the core breeding areas of these two species in our region. We have now remarked in the revised version the representativeness of our study area in the European context to help the audience (including researchers working with these species) understand the relevance of our results for other Atlantic Mountain heathlands (lines 140 and 144-145); and two new references (highlighted in yellow):
Rego, P. R.; Guitián, M. A. R.; Castro, H. L.; Da Costa, J. F.; Sobrino, C. M. Loss of european dry heaths in NW Spain: A case study. Diversity 2013, 5, 557–580.
Fagúndez, J. Heathlands confronting global change: Drivers of biodiversity loss from past to future scenarios. Ann. Bot. 2013, 111, 151–172.
In addition, we think that the fine-scale habitat characterization based on Landsat images (30-m resolution) allows to model these species with confidence. In fact, land-use changes and their impacts on habitat quality predicted by our models are in line with other studies carried out in our region at larger scale. See our previous studies:
Regos, A.; Tapia, L.; Gil-Carrera, A.; Domínguez, J. Monitoring protected areas from space: A multi-temporal assessment using raptors as biodiversity surrogates. PLoS One 2017, 12, e0181769.
Tapia, L.; Dominguez, J.; Rodriguez, L. Modelling habitat selection and distribution of Hen harrier (Circus cyaneus) and Montagu’s harrier (Circus pygargus) in a mountainous area in Galicia. J. Raptor Res. 2004, 38, 133–140. 441
Tapia, L.; Domínguez, J.; Rodríguez, L. Hunting habitat preferences of raptors in a mountainous area (northwestern Spain). Polish J. Ecol. 2008, 56, 323–333.
Tapia, L.; Regos, A.; Gil-Carrera, A.; Dominguez, J. Assessing the temporal transferability of raptor distribution models: Implications for conservation. Bird Conserv. Int. 2018, 28, 375–389. 450
Tapia, L.; Regos, A.; Gil-Carrera, A.; Domínguez, J. Unravelling the response of diurnal raptors to land use change in a highly dynamic landscape in northwestern Spain: an approach based on satellite earth observation data. Eur. J. Wildl. Res. 2017, 63, 40.
(2) A further emerging problem may be that although breeding Hen Harriers predominately hunt within 1-2 km from their nest, they can travel up to 9 km from their nests and can have an average home-range of 8 km2 (Arroyo et al. 2014, Bird Study 61/1). Hence, it is very likely that the same birds were counted at various census points. Again, then pseudoreplication / spatial autocorrelation may severely weaken the reliability of the models.
Response: The observers (who are also co-authors of the work) have a very long experience working with raptors and with these species in our study area. They were aware of the risk of pseudoreplication but, as commented above, every contact within a transect was attribute to one individual, the risk of the same individual was counted twice within the same transect was very low. The replication of the transect over time ensure that all absences recorded are real and not pseudo-absences (critical issue in habitat suitability modelling). The number of individuals recorded in each transect is low, which make them easily controllable during the survey. We are therefore confident that pseudo-replication and spatial autocorrelation was not a problem, as that risk was carefully considered by observers when sampling the populations. However, we have included in the discussion a few lines to make the readers aware of the risk associated with these issues in the case of species with large home ranges such as these raptors, in particular the large hunting distances of breeding male and female Hen Harriers (lines 370-374). We have also included the reference suggested by the reviewer to support these aspects/limitations.
(3) In Montagu’s harrier the situation may be even more problematic since this species is semi-colonial (e.g. Arroyo et al. 2004, BWP Update 6: 41-55). This may explain why predictions of habitat suitability based on occurrence data do not result in close correlations with spatio-temporal abundance changes.
Response: thanks very much for this pertinent comment and the suggested reference (now included). The semi-colonial behavior of the Montagu’s harrier can be another factor that could explain why predictions of habitat suitability based on occurrence are not strongly correlated with spatiotemporal changes in abundance. This is in fact an issue that has not been mentioned in previous metanalysis about the use of habitat suitability models as proxy for abundance, and autoecology of our target species can provide insights into this hotly debated relationship (see metanalysis of Weber et al. 2017). We add a few lines in this regard to enrich the discussion and contribute to the debate (lines 370 and 374) together with recommendations to overcome these limitations (lines 463-469).
At least it has to be comprehensively addressed in the discussion to what extent these two issues (2) and (3) may affect the performance of the calculated models!
Response: In the revised version, we have now explicitly addressed these issues particularly important for migratory, semi-colonial species with large home range. We think that this cannot only improve our work but also call the attention to important issues to be considered in future studies when using habitat suitability models to infer spatiotemporal changes in abundance, especially with raptors (lines 370 and 374; line 441, and lines 463-469).
(4) Surveys were conducted between 1 April and 31 August. Since April is the main spring migration period of Montagu’s harrier many migrants may pass the area, which may utilize different habitats than breeding birds. Combining both types of birds – breeding and migrating individuals – may affect the predicted habitat suitability. This is an issue, which should be also clarified.
Response: Sorry, it was my mistake, my colleagues confirmed that the starting date of the fieldwork and the surveys started from the second half of April. Anyway, regarding the migratory behavior of the Montagu’s harrier and the dispersive character of the Hen harrier, these populations are very philopatric, that is, individuals tend to return to breed in this area every year (sometimes even in the same shrub patch). In addition, the fledglings stay in the breeding area until September (we did not observe any flocks of individuals assembling before migration, i.e., no migratory restlessness in August). We now add a few lines in the survey design section to correct the dates of both surveys, between 15th Abril and 31st August (in 1997 and 2017), and explain the philopatric behavior of these species, which strongly reduce the risk of recording individuals going northern areas (lines 156 and 160).
Specific comments
It is unnecessary in the text of the discussion to refer to specific figures again in parentheses.
Response: We removed any reference to specific figures in the discussion.
Figure 2: I fear that in this black-and-white figure not everything can be seen that is necessary for the interpretation of the results. Please check.
Response: All figures are in color. Figure 2 should show the relative importance of each predictor variable. We have just realized that figures 2, 4 and 5 are wrong in the PDF (not in the Word file). It might be caused by the automatic conversion from Word to PDF during the submission process. We apologies for this error. We will ask the editorial team for the possibility of uploading the figures as separate files to ensure a correct visualization.
Here all original figures:
https://nubeusc-my.sharepoint.com/:f:/g/personal/adrian_regos_usc_es/Eg2jXBBLDMROt4kOA8BvzxYB7xq6QwzQGQ2EOiYvt9SNSg?e=A0hKxG
Editorial team: Please ensure that the revised version include the figures with colors. Thank you!
Reviewer#2
The paper entitled “Caution is needed when using niche models to infer changes in species abundance: the case of two sympatric raptor populations” submitted to Animals deals with habitat loss affecting populations of two sympatric harrier species in Spain.
The MS is well-written and I only have some minor revisions before acceptance.
Response: thank you very much for your time and positive feedback on our work. We have addressed all your comments to improve our work.
- Lines 46-47. I guess it is the most important cause of biodiversity loss at all.
Response: Yes, agree. There are several global assessments that point out to habitat loss as the most important threat (see e.g., Diaz et al 2019). We now state that habitat loss is the most important cause of biodiversity loss (line 54).
Reference:
Díaz, S.; et al. Summary for policymakers of the global assessment report on biodiversity and ecosystem services of the Intergovernmental Science-Policy Platform 409 on Biodiversity and Ecosystem Services; IPBES secretariat: Bonn, Germany, 2019.
- Line 57. Not only environmental covariates, also climatic ones!
Response: we now clarify that the environmental covariates are mostly represented by climatic and topographic variables, being widely used in niche modelling due to their large availability despite the relevance of local habitat variables (lines 64-65).
- Lines 100-110. It is not exactly clear to me why authors selected exactly those species and why it is so important to study these harrier species. Authors say that they are affected by habitat loss, but how? At lines 97-99, you talk about decline. How much are they declining and in which time?
Response: Previous studies developed by our team showed important declines for these open-habitat raptors in the province, where the study area is located, over the last 20 years. These changes in habitat availability and quality were caused by land-use conversions from heathlands and grasslands to fast-growing tree plantations (mostly Pinus but also Eucalyptus spp.) and native forest expansion due to rural abandonment processes (loss of traditional agricultural and livestock practices) and intensive forest practices. We now added this complementary information to help readers understand the suitability of these two species as case study, in addition to their relevance in terms of conservation of mountain heathlands and other species linked to these habitats (see lines 113-118).
References:
Regos, A.; Tapia, L.; Gil-Carrera, A.; Domínguez, J. Monitoring protected areas from space: A multi-temporal assessment using raptors as biodiversity surrogates. PLoS One 2017, 12, e0181769.
Tapia, L.; Dominguez, J.; Rodriguez, L. Modelling habitat selection and distribution of Hen harrier (Circus cyaneus) and Montagu’s harrier (Circus pygargus) in a mountainous area in Galicia. J. Raptor Res. 2004, 38, 133–140. 441
Tapia, L.; Domínguez, J.; Rodríguez, L. Hunting habitat preferences of raptors in a mountainous area (northwestern Spain). Polish 442 J. Ecol. 2008, 56, 323–333.
Tapia, L.; Regos, A.; Gil-Carrera, A.; Dominguez, J. Assessing the temporal transferability of raptor distribution models: Implications for conservation. Bird Conserv. Int. 2018, 28, 375–389. 450
Tapia, L.; Regos, A.; Gil-Carrera, A.; Domínguez, J. Unravelling the response of diurnal raptors to land use change in a highly dynamic landscape in northwestern Spain: an approach based on satellite earth observation data. Eur. J. Wildl. Res. 2017, 63, 40.
Regos, A., Arenas-Castro, S., Tapia, L., Domínguez, J., & Honrado, J. P. (2021). Using remotely sensed indicators of primary productivity to improve prioritization of conservation areas for top predators. Ecological Indicators, 125, 107503.
- Lines 105-110. Authors should include some predictions.
Response: We agree with the reviewer that the last part of the introduction also benefit from including some predictions or expectations regarding our hypotheses and analysis. However, our predictions were already stated in lines 93-97: “we would expect model-predicted environmental suitability to be significantly correlated with species abundance, thus supporting the use of SRS-based ENMs to estimate species abundance. If so, both the spatial and temporal variation in species abundance should be explained (at least partly) by changes in environmental suitability predicted by these models”.
- Methods are clearly stated, and replicable.
Response: Thank you!
- I totally do not understand Figure 2, it is almost unreadable and provide me with no information.
Response: Figure 2 should show the relative importance of each predictor variable. We have just realized that figures 2, 4 and 5 were wrong in the PDF (not in the Word file). It might be caused by the automatic conversion from Word to PDF during the submission process. We apologize for this error. We will ask editorial team for the possibility of uploading the figures as separate files to ensure a correct visualization.
Anyway, here you can find all original figures:
https://nubeusc-my.sharepoint.com/:f:/g/personal/adrian_regos_usc_es/Eg2jXBBLDMROt4kOA8BvzxYB7xq6QwzQGQ2EOiYvt9SNSg?e=A0hKxG
- Line 289. Delete “study” before “findings”.
Response: done.
- Line 295. You say “various authors” but you cited only one work, please increase literature search or change “various authors” with “one study”.
Response: We only cited one work because that study is a metanalysis about that topic. We now encourage readers to see “reference therein” and include another key reference.
Line 321. Delete “the” before “findings”.
Response: done.
